# Does GPA matter for university graduates' wages? New evidence revisited

Tao Zou[1]*, Yue Zhang[2], Bo Zhou[3]

**1** China Research Center for Educational Outcomes, Southwestern University of Finance and Economics, Chengdu, Sichuan, China, **2** Research Institute of Economics and Management, Southwestern University of Finance and Economics, Chengdu, Sichuan, China, **3** Department of Public Finance and Taxation, University of International Business and Economics, Beijing, China

* zoutao120240@swufe.edu.cn

**Data Availability Statement:** All relevant data are within the paper and its Supporting Information files.

**Funding:** The author(s) received no specific funding for this work.

## Abstract

This paper examines the effect of GPA on graduating students' wages using a data set from an elite university in China. Students are homogenous since their majors are closely related to economics and business The OLS regression results indicate that GPA has positive and significant impacts on wages on average. As GPA increases by 1 unit, the starting monthly wage increases by 29.6 percent on average, and the wage in the survey year that is 3–5 years after graduation (current wage) soars by 25 percent. Theoretically, the GPA matters for the wages due to both the human capital or signaling effect. Given that the signaling effect should diminish over time, and the effect on starting wage is a little larger than that on current wage, it is suggested that signaling effect of GPA should be trivial, and high GPA is associated with high wage should be mainly due to the human capital effect. These results are robust to different model specifications. The distributional analysis suggest that the effects are positive for both wages and significant for almost all quantiles. In addition, the effect is basically the same from the 0.05th to 0.80th quantiles, and then rises as the starting wage increases. The effect on current wage is a U shape from the 0.05th to 0.60th quantile, and then becomes an inverse-U shape with peaks at the 0.75th and 0.80th quantiles where the effect is 82.2 percent when GPA increases by one unit.

## Introduction

It is reported by *People's Daily*, an online news and comments aggregator, 2013 was the most difficult employment year in China's history [1]. The total number of college graduates nation-wide reached 6.99 million, the largest number since 1997 [2]. At the same time, job vacancies declined substantially [3]. This situation has not improved. From *Research Report on College Students with Employment Difficulties*, in 2020, approximately 5.9 million new graduates entered the labour market, while in June 2020, 26.3% of the 2020 fresh graduates were still seeking jobs [4]. This means that college students are about to face a more severe employment challenge. In this situation, the employment problem of college graduates has become a serious concern in society as a whole. What skills and knowledge should universities students obtain

**Competing interests:** The authors have declared that no competing interests exist.

to get greater advantages in the labour market? There is a hot debate on this: some people think that it is better for college students to study hard and strive to improve their grade point average (GPA), while others suggest that college students should participate more in club activities and look for opportunities to intern in order to develop practical nonacademic skills.

There are relatively few studies on the impact of nonacademic performance on income in the existing literature. One type of literature focuses on participation in sports activities [5,6]. Another stream of literature focuses on university club activities [7–9]. The basic conclusion is that the experience of nonacademic performance can significantly improve the salary of university graduates.

In contrast, there is no consensus on the impact of academic performance in the research. Theoretically, the impact on income is mainly through the human capital effect [10] and signal effect [11]. The human capital effect suggests that higher academic performance such as obtaining an education degree leads to greater personal productivity, thereby increasing the income of workers. The signal effect, also called sheepskin effect in the literature, believes that academic performance as a signal of labour productivity can help distinguish it from employees' productivity. When individuals work in particular firms longer and longer, the employers can directly observe the true difference in productivity among workers. As a result, the signal effect of academic performance should diminish over time; in contrast, if the academic performance matters mainly due to the human capital effect, the effect of academic performance should not reduce substantially and even the effect can increase over time [12,13].

Empirically, GPA is often used as a measurement of students' academic performance. It is believed that GPA not only reflects the cognitive ability of students [14,15] but is also related to some noncognitive abilities. It represents conscientiousness, academic discipline, and successful leadership experience since a high GPA requires persistence in learning over time [16–18]. Previous studies exploring the impact of GPA on students' wages have different results. Most studies show that the GPA of undergraduate students has a direct and positive impact on the earnings of undergraduates [15,19,20]. However, other studies show that the GPA has no effect on income for some groups [21,22]. Previous studies often use samples from various universities. It is possible that the effect of GPA also reflects some unobserved differences among universities' educational standards and among different majors.

This paper aims to investigate the impact of GPA on the wages of university graduates. All graduates are from an elite university in China, and nearly all of the students' majors are related to economics and finance. Thus, on the one hand, our data have some limitations and may not represent the whole population of university graduates. On the other hand, all people are very homogenous in terms of professional skills and knowledge and even similar in terms of personality traits. In addition, we use administrative data that strictly record all kinds of performance of students in school, which allows us to estimate the role of GPA more accurately. Specifically, we try to answer the following three questions: (1) Does GPA affect university graduates' wages? (2) Does the impact of GPA vary between starting wages and wages 3–5 years after graduation? (3) Is the effect heterogeneous over the wage distribution?

Using the data of 706 graduates who entered the university in 2009 and 2010, the OLS regression analyses suggest a positive and significant relationship between GPA and wages: when GPA increases by 1 unit, the starting monthly wage increases by 0.259 log points (29.6 percent), and the wage in the survey year (2018), which was 3–5 years after graduation ("current wage" hereafter), increases by 0.233 log points (26.2 percent). These results are robust to controlling for different fixed effects and the functional form of GPA. The unconditional quantile regression results show that the positive effect of GPA on wages is almost significant in all quantiles. For starting wages, the effect is basically the same between the 0.05th and 0.80th quantiles, and it then rises as wages increase. The effect on current wage is a U shape from the

$0.05^{th}$ to $0.60^{th}$ quantile, and then becomes an inverse-U shape with peaks at the $0.75^{th}$ and $0.80^{th}$ quantiles where the effect is 82.2 percent when GPA increases by one unit. In summary, the comparison between starting and current wages suggests that a higher GPA, as an indicator of human capital, mainly results in higher wages, while the signal effect in students' first job should be trivial.

Our paper makes several contributions to the literature. First, to the best of our knowledge, our paper is the first study about the relationship between GPA and the labour market outcomes of university graduates in China. There have been many studies about the effects of nonacademic activities or awards and university quality on labour market outcomes in China, but for some unknown reason, the role of academic performance has been largely ignored. This paper bridges this gap in the literature. Second, there have been many studies about the human capital and signal effects of higher education. The research mainly focuses on the effects of the degree or the university rank. Our paper provides another dimension to this research, which can help deepen the understanding of the role of academic performance in higher education. Third, the paper provides clear guidance for university students, at least for students in economics- and business-related majors. The findings in our paper imply that university students still need to pay attention to academic performance, which has a positive effect on their future wages.

The remainder of the paper is structured as follows. Section 2 describes our data and presents our empirical model. Section 3 presents the results, including the basic results, robustness checks, and distributional effects of GPA on wages. Finally, Section 4 summarizes and concludes the paper.

## Materials and methods

### Data

In this paper, we investigate the effect of GPA on the labour market outcomes of graduates from an elite university in China. There were two higher education programs: Project 211 for elite universities and Project 985 for top universities. Project 211 includes 116 universities, and 39 of these 116 universities are also Project-985 universities. This study exploits two data sources: questionnaire survey data among graduates, and university administrative data that include all of the basic demographic information and the academic achievements of these graduates. In the data we accessed and used for analyses, all of the students' identities and personal information were anonymized.

The questionnaire survey was conducted in 2018. The university randomly sent an online questionnaire to 1,000 graduates who entered university in 2009 and 2010. The university has a roaster of graduates, which includes some "permanent" contact information, such as email and QQ account (a popular instant massage application in China). 1000 graduates were randomly selected from this roaster. The most and least successful graduates are less likely to respond the survey, because they are either too busy to fill in the survey form or reluctant to report their 'frustrated' situation. This may cause some bias, but we cannot investigate more due to the data limitation. Finally, 706 effective questionnaires were collected. The questionnaire survey includes two parts: one on labour market outcomes when students initially entered the labour market, and the other on labour market outcomes in 2018, 3 to 5 years after graduation. The university administration database contains students' basic demographic information, family background, academic performance (GPA), and some other campus performance indicators. For instance, students' comprehensive quality information includes the performance of students in scientific research ability, innovation ability, ideological and moral accomplishment, artistic and physical accomplishment, and social practice ability.

This university we study is a finance and economics university. The majors of most students are related to finance, economics and business. After graduation, they highly concentrated in some particular sectors compared to other comprehensive universities. This feature is a two-edged sword: (1) many observations are made in the same or similar majors and works; thus, the estimated results are more precise; but (2) because there are no additional observations in other majors and works, we know little about the extent to which our results can be extended.

## Ethical approval

This research has gone through two ethical reviews for each of our related research projects by Ethics Committee of Student Career Planning and Guidance Center, Southwestern University of Finance and Economics (SWUFE).

> March 6, 2018
>
> To whom it may concern,
>
> It is hereby certified that, a) the SWUFE Graduates' labour Market Outcome Survey has been reviewed, approved and granted to study, b) This project does not involve ethical relevant information.
>
> Ethics Committee of Student Career Planning and Guidance Center
>
> Southwestern University of Finance and Economics

and

> June 24, 2020
>
> To whom it may concern,
>
> It is hereby certified that, a) this research 'Does high GPA predict or causes high wage? New evidence revisited' has been reviewed, the data in this study has been data masking, and students' identity and personal information has been anonymized.
>
> Ethics Committee of Student Career Planning and Guidance Center
>
> Southwestern University of Finance and Economics

## Summary statistics

Table 1 presents the summary statistics. We use two variables to measure graduates' labour market outcomes: the starting monthly wage of the first job (starting wage, hereafter) and the monthly wage at the time of the survey namely, 3–5 years after graduation (current wage, hereafter). The starting wage can be viewed as the payoff offered by employers when they do not have complete information on the graduates' abilities, while the current wage is the payoff when employers know more about the graduates' abilities. Table 1 shows that, on average, the starting wage is approximately 7,777 yuan, while the current wage is approximately 11,954 yuan. The wages are deflated to the price level of 2018 using the CPI.

GPA, as a measure of academic performance, reflects the cognitive and noncognitive abilities of students. Students often would like to report their GPA in the resume they use when they are looking for their first job. The GPA range is 0–5, and the comparison between the percentile system and GPA is 60 to 100 points when GPA is 1.0 to 5.0, and 0 corresponds to less than 60 points. From Table 1, we can see that the mean GPA is 3.275 and the standard deviation is 0.385. With respect to nonacademic abilities, we consider a dummy variable indicating whether

an individual obtained a nonacademic scholarship in the university. The nonacademic award refers to innovation and entrepreneurship, scientific research, ethics, or practical awards. If students obtain one of them, then the dummy variable equals 1 and vice versa. Table 1 shows that almost half of the students in the sample have received a nonacademic scholarship.

Table 1 also shows that majority of the students work in the finance industry, and the number of students in other industries is quite small. Thus, we aggregate all industries other than

**Table 1. Summary statistics.**

| Variables | Mean | SD |
|---|---|---|
| **Wage** | | |
| Starting monthly wage (yuan) | 7776.546 | 4922.777 |
| Current monthly wage (yuan) | 11954.392 | 8360.492 |
| ln(starting monthly wage) | 8.820 | 0.510 |
| ln(current monthly wage) | 9.224 | 0.557 |
| **Performance in university** | | |
| GPA | 3.275 | 0.385 |
| Obtain a nonacademic scholarship | 0.479 | 0.500 |
| **Individual and employment characteristics** | | |
| Male | 0.415 | 0.493 |
| Status after graduation | | |
| *Directly employed after graduation* | 0.574 | 0.495 |
| *Postgraduate study after graduation* | 0.363 | 0.481 |
| *Job-waiting after graduation* | 0.064 | 0.244 |
| Major | | |
| *Economics* | 0.017 | 0.129 |
| *Finance* | 0.329 | 0.470 |
| *Management* | 0.460 | 0.499 |
| *Math and engineering* | 0.081 | 0.273 |
| *Arts, humanities and other social sciences* | 0.113 | 0.317 |
| Industry | | |
| *Finance* | 0.560 | 0.497 |
| *Others* | 0.440 | 0.497 |
| Employer type | | |
| *Public sector* | 0.163 | 0.369 |
| *State-owned Enterprise* | 0.451 | 0.498 |
| *Others* | 0.387 | 0.487 |
| **Family background** | | |
| Economic status: *Rich* | 0.581 | 0.494 |
| Parental education | | |
| *Junior middle school and below* | 0.217 | 0.412 |
| *Senior middle school* | 0.286 | 0.452 |
| *College and above* | 0.497 | 0.500 |
| Parental occupation | | |
| *Unemployed or retired* | 0.127 | 0.334 |
| *Professional* | 0.242 | 0.429 |
| *Management* | 0.293 | 0.456 |
| *Peasant or rural-urban migrant worker* | 0.174 | 0.380 |
| *Local urban worker* | 0.163 | 0.370 |
| Observations | 706 | |

finance together, which consists of 44% of the sample. Most of the students chose jobs in state-owned enterprises (SOEs) (45%), private-owned enterprises (39%), and finally, the public sector (16%).

The family income level was originally classified into four levels: (1) very poor, (2) poor, (3) rich, and (4) very rich, and students were asked to evaluate their family's economic status. However, there are only very few people who choose very poor and very rich. Therefore, we combine (1) and (2) together as "poor" and combine (3) and (4) together as "rich". Table 1 shows that nearly 60% of the students believe that their family is "rich". This is consistent with the finding that it becomes increasingly difficult for children from "poor" families to enter elite universities. In addition, Table 1 also illustrates that approximately half of the parents of the students in this university have a college education or higher and are more likely to have a professional or management job. This confirms the subjective evaluation of family economic status.

In addition to the summary statistics presented in Table 1, we also plot the unconditional relationship between GPA and the log of wages (Fig 1). It is clear that GPA is positively related to both the starting monthly wage and the current wage. However, the slope for the current wage is slightly lower than the starting wage. This seemingly tells us that GPA has a signaling effect on wages; but given that the effect of GPA decreases slightly, the signaling effect should not be large.

## Empirical model

To estimate the effect of students' GPAs in university on labour market performance, we consider the following empirical model:

$$Y_{igpw} = \alpha GPA_{igpw} + \delta' X_{igpw} + u_g + u_p + u_w + \varepsilon_{igpw} \tag{1}$$

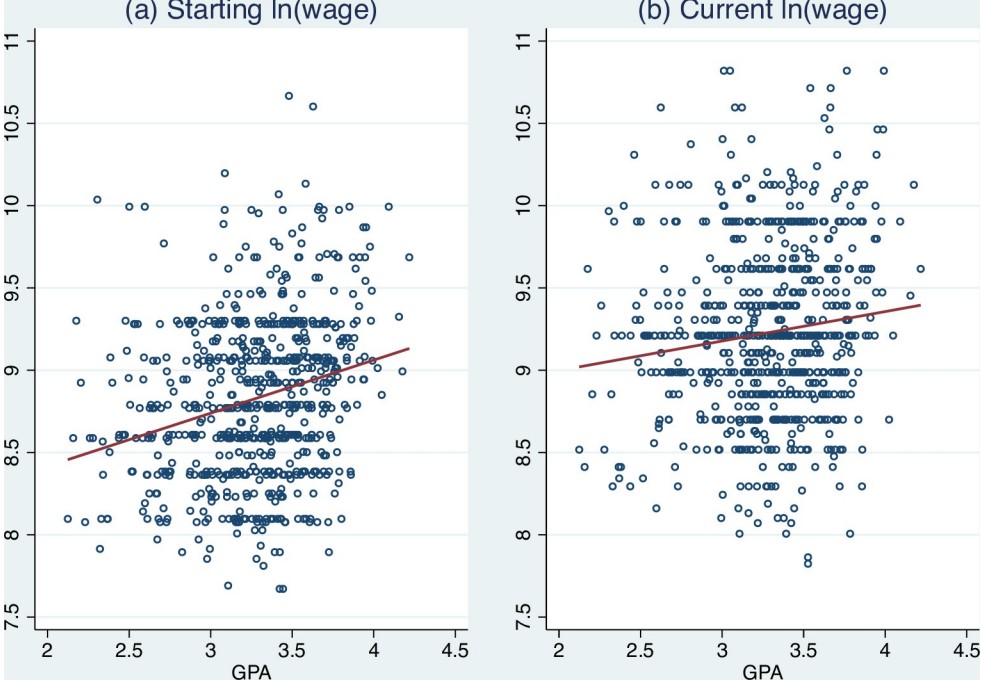

**Fig 1. The unconditional relationship between GPA and log of wage.**

where the subscript *igpw* denotes a graduate *i* who entered university in year *g* (grade, hereafter), took the college entrance examination in province *p* (home province, hereafter), and was working in province *w*(residential province, hereafter) in 2018, the survey year; *Y* is a labour market outcome, the graduate's starting monthly wage or the current monthly wage, both of which are taken as logs in the regression.

In this model, GPA is the primary variable of interest; it measures academic performance in universities, and thus, $\alpha$ measures the effect of academic performance on labour market outcomes. $X_{igpw}$ is a vector of control variables. First, we include a dummy variable indicating whether the graduate obtained a nonacademic scholarship in $X_{igpw}$. This is because academic performance is likely to be correlated with nonacademic performance in universities. Including this variable is helpful to isolate the effects of academic performance from nonacademic performance in universities. In addition, we also control for gender, parents' occupation, major, his or her choice after graduation, and industry fixed effects.

Next, we control for a set of fixed effects: grade fixed effects ($u_g$), home province fixed effect ($u_p$) and residential province fixed effect ($u_w$). Grade fixed effects and home province fixed effects can account for differences in average ability in a particular grade cohort and particular province cohort. Note that in China, admission to a higher education institution is organized at the province level; thus, the admission threshold varies from province to province, and the threshold can also change over time. Residential province fixed effects can help eliminate the influence of different characteristics of the local labour market. Finally, $\varepsilon_{igpw}$ is the idiosyncratic error term, and standard errors are clustered at the school level throughout.

## Results

### Baseline results

Table 2 presents the baseline results of Eq (1) for the starting monthly wage (Columns 1–3) and their current wage (Columns 4–6). In all Columns, we include the full set of control variables including the grade fixed effects. In Columns (1) and (4), we control for the home provincial fixed effects, which can help eliminate the sorting effect in each province because the national college entrance exam (NCEE) is organized by each province. It is shown that one additional GPA point can increase the starting wage by 0.259 log points (29.6 percent) and increase the current wage by 0.278 log points (32.0 percent), and they are both statistically significant at the 1% level. These correspond to 29.6% and 32% increments in the starting wage and current wage, respectively. This is a huge effect. It is found that compared to the three-year college and senior high school graduates, the premium of a four-year university graduate was only 0.336–0.436 log points (39.94–54.65 percent) in 2005 [22]. Although we study the labour market 13 years later than this study, our results are still impressive in the sense that the wage variation among those with the same degree who even graduated from the same university can still be large, and the GPA is valuable even though it might not have such large premia as a difference in degree.

In Columns (2) and (5), we control for the current residential provincial fixed effects but not the home provincial fixed effects. It is well known that the spatial income disparity in China is quite large, such as between Shanghai and Shanxi. Controlling for these residential provincial fixed effects can help eliminate local labour market differences. Compared to Column (1), the result in Column (2) is the nearly the same; however, compared to Column (4), the effect of GPA presented in in Column (5) reduces slightly more, from 0.278 log points (32.05 percent) to 0.225 log points (25.23 percent), but it is still significant at the 1% level.

In Columns (3) and (6), we control for both the home provincial and currently residential provincial fixed effects–this is our most preferred model specification. The result in Column

**Table 2. The results of GPA on the starting wage and current wage.**

| VARIABLES | Log of starting monthly wage | | | Log of current monthly wage | | |
|---|---|---|---|---|---|---|
| | (1) | (2) | (3) | (4) | (5) | (6) |
| GPA | 0.259*** | 0.254*** | 0.259*** | 0.278*** | 0.225*** | 0.233*** |
| | (0.047) | (0.052) | (0.046) | (0.060) | (0.056) | (0.064) |
| Obtain a non-academic award | -0.004 | -0.022 | -0.018 | 0.097 | 0.084 | 0.067 |
| | (0.051) | (0.047) | (0.047) | (0.058) | (0.052) | (0.055) |
| **Individual and employment characteristics** | | | | | | |
| Male | 0.170*** | 0.145*** | 0.128*** | 0.296*** | 0.229*** | 0.217*** |
| | (0.038) | (0.034) | (0.039) | (0.053) | (0.052) | (0.055) |
| Postgraduate study after graduation | 0.294*** | 0.251*** | 0.250*** | 0.027 | -0.085 | -0.076 |
| | (0.052) | (0.049) | (0.061) | (0.043) | (0.059) | (0.071) |
| Job-waiting after graduation | 0.101 | 0.078 | 0.085 | -0.032 | -0.043 | -0.041 |
| | (0.084) | (0.097) | (0.098) | (0.041) | (0.040) | (0.039) |
| Major: Finance | 0.125** | 0.074 | 0.089* | -0.140*** | -0.095* | -0.098 |
| | (0.044) | (0.048) | (0.049) | (0.038) | (0.046) | (0.058) |
| Major: Management | 0.116** | 0.074 | 0.091* | -0.102** | -0.076* | -0.059 |
| | (0.048) | (0.051) | (0.050) | (0.047) | (0.041) | (0.058) |
| Major: Mathematical sciences | 0.177*** | 0.129** | 0.154*** | 0.022 | 0.029 | 0.051 |
| | (0.053) | (0.057) | (0.045) | (0.029) | (0.030) | (0.055) |
| Major: Arts, humanities and other social sciences | -0.009 | -0.030 | -0.025 | -0.140** | -0.126*** | -0.104* |
| | (0.035) | (0.042) | (0.040) | (0.047) | (0.041) | (0.058) |
| Employer type: SOE | 0.100* | 0.049 | 0.044 | 0.340*** | 0.236*** | 0.242*** |
| | (0.056) | (0.062) | (0.061) | (0.067) | (0.061) | (0.067) |
| Employer type: Others | 0.051 | -0.023 | -0.036 | 0.450*** | 0.274** | 0.272** |
| | (0.053) | (0.051) | (0.054) | (0.093) | (0.094) | (0.109) |
| Industry: Finance | 0.148** | 0.147*** | 0.142*** | 0.139 | 0.118* | 0.120* |
| | (0.053) | (0.048) | (0.044) | (0.081) | (0.067) | (0.065) |
| **Family background** | | | | | | |
| Family economic status: Rich | 0.119*** | 0.104*** | 0.103** | 0.164*** | 0.134*** | 0.146*** |
| | (0.037) | (0.035) | (0.036) | (0.028) | (0.028) | (0.027) |
| Parents' occupation: Professional | 0.036 | -0.008 | -0.001 | 0.045 | 0.039 | 0.012 |
| | (0.035) | (0.044) | (0.044) | (0.081) | (0.083) | (0.092) |
| Parents' occupation: Management | 0.080* | 0.027 | 0.033 | -0.019 | -0.052 | -0.067 |
| | (0.039) | (0.029) | (0.021) | (0.064) | (0.055) | (0.055) |
| Parents' occupation: Peasant or migrant workers | -0.038 | -0.041 | -0.053 | -0.017 | 0.000 | -0.016 |
| | (0.061) | (0.059) | (0.062) | (0.063) | (0.078) | (0.080) |
| Parents' occupation: Local urban worker | 0.021 | 0.022 | -0.004 | 0.016 | 0.014 | -0.007 |
| | (0.069) | (0.074) | (0.080) | (0.052) | (0.056) | (0.062) |
| Parental education: Senior middle school | 0.033 | 0.022 | 0.020 | 0.012 | -0.012 | -0.014 |
| | (0.053) | (0.041) | (0.053) | (0.054) | (0.053) | (0.053) |
| Parental education: college and above | 0.063 | 0.073 | 0.063 | 0.101* | 0.100* | 0.108** |
| | (0.066) | (0.053) | (0.055) | (0.049) | (0.053) | (0.050) |
| Constant | 7.399*** | 7.575*** | 7.565*** | 7.685*** | 8.064*** | 8.042*** |
| | (0.139) | (0.177) | (0.157) | (0.196) | (0.155) | (0.182) |
| Grade FE | Yes | Yes | Yes | Yes | Yes | Yes |
| Home province FE | Yes | No | Yes | Yes | No | Yes |
| Residential province FE | No | Yes | Yes | No | Yes | Yes |
| Observations | 686 | 645 | 645 | 686 | 645 | 645 |

*(Continued)*

**Table 2.** (Continued)

|  | Log of starting monthly wage | | | Log of current monthly wage | | |
| --- | --- | --- | --- | --- | --- | --- |
| **VARIABLES** | **(1)** | **(2)** | **(3)** | **(4)** | **(5)** | **(6)** |
| R-squared | 0.346 | 0.365 | 0.395 | 0.308 | 0.398 | 0.424 |

Note: Non-academic award includes innovation and entrepreneurship, scientific research, ethic, or practical award. In all regressions, the omitted status after graduation is "*Directly employed*"; the omitted major is "*Economics*"; the omitted industry is "*others*"; the omitted employer type is "*Public sector*"; the omitted parental education is "*Junior middle school or below*"; the omitted parental occupation is "*unemployed or retired*". Robust Standard errors are in parentheses, which are clustered at the school level.

\*\*\* p<0.01,

\*\* p<0.05,

\* p<0.1.

(3) is the same as in Column (1), and the result in Column (6) is between the Columns (4) and (5)–one additional GPA implies 0.233 log points (26.2 percent) increase in the current monthly wage.

Based on these results, it is clearly illustrated that GPA has a positive effect both on the starting wage and the current wage. How can these results be interpreted? It is well known that GPA represents students' ability and quality and reflects human capital acquisition, and GPA can also have a signal effect (sheepskin effect) [14,18,23]. For human capital, GPA represents human capital acquired in college, so higher GPA means higher human capital and employees will probably have higher job productivity. For signal effect, GPA plays the informational role of differentiating ability levels of prospective employees. The signal effect can be expressed as: "Indeed, one of the crucial roles of educational screening is presumably to allow employers to select the more talented for jobs which involve considerable on-the-job training [24]." This implies that the students initially enter the labour market, and both the human capital effect and the signal effect of GPA may exist. In the recruitment process, the HR staff may find that job candidates with high GPAs truly have high abilities, but it is also possible that the HR staff feel that job candidates have higher abilities and qualities and, thus, offer a higher starting wage. Theoretically, however, we cannot distinguish these two different effects. The current wage is the labour market outcome when people have worked 3–5 years following graduation. At this time point, employers have more information about people's abilities, and they no longer need to rely on GPAs. That is, the effect of GPA on wages should mainly affect the impact of human capital. Comparing the results for the starting and current wages, we can conclude that GPA matters to the labour market mainly through the human capital effect rather than the signal effect.

It is interesting that nonacademic awards do not have a significant effect on wages. This may be because nonacademic awards in economics- and business-related majors or at this particular university are not informative to employers, or at least nonacademic awards do not help accessing the first job. The starting wage of males is approximately 0.13 log points (13.9 percent) higher than that of females, while the premium of males becomes larger for the current wage. To some extent, this result is consistent with one research which uses a firm-level dataset in China's private sector [25]. This change in gender discrimination over age may be due to the motherhood penalty [26]. The starting wage is much higher with a postgraduate degree than without it; however, this does affect the current wage. This result suggests that the postgraduate degree has no excess return compared to a bachelor's degree in the long run; this also suggests that we need to evaluate whether there is an overeducation problem, or at least an overinvestment in postgraduate education, in China.

It is interesting that the academic major is very important for the starting wage. For example, compared to economics majors, mathematical sciences, finance and management majors are more likely to have higher starting wages, although it is only statistically significant at the 1% level for the first major. However, in the long run, there is no significant difference between the wages of these majors. Our survey is based on an economic and finance university of Project 211, and the jobs may not be so different for majors in other areas of knowledge. This may explain the insignificant difference among majors in wages in the long run. Working in the finance industry has a similar effect: the finance industry has a significant premium in the starting wage, while it becomes only marginally significant for the current wage. Compared to the public sector, jobs in the SOE and private sector have no difference in the starting wage but pay approximately 0.25 log points (28.4 percent) more than in the public sector, which is consistent with our expectations.

With respect to family background, we find that parental occupation has no significant effect on either starting or current wages. However, the people from rich families earn more than those from poor families, and this effect is larger for the current wage than for the starting wage. The effect of parental education has a similar pattern: parents with a bachelor's degree or higher increase the current wage by approximately 0.10 log points (10.5 percent), although it is not significant for the starting wage. These results confirm the findings that both human capital and wealth contribute to intergenerational income transmission in China, but wealth contributes more than human capital [27].

## Robustness checks

In this section, we conduct three groups of robustness checks. Table 3 presents these regression results, in which Panels A and B are for the starting wage and current wage, respectively.

First, we try to control for different fixed effects, which is more flexible than the baseline model. These results are presented in Columns (1)-(3). In Column (1), we still control for the home provincial fixed effects, but we also control for the grade-currently residential provincial fixed effects. Compared to the baseline model, with this model specification, we can take account of particular province-years' labour markets. The results with this specification remain similar to the baseline model's results. In Column (2), we control for current provincial fixed and control for the grade-home provincial fixed effects. Using this model specification, we can consider the sorting effect in the university admission process. This is stricter than the baseline model because the students' abilities in different years but from the same province might be different. Similar to the first exercise, the results remain similar to the baseline results. In Column (3), we control for both the grade-home province fixed effects and the grade-currently residential provincial fixed effects. Again, this model specification has similar results as the baseline model. That is, our results are robust to the way in which we control for the fixed effects.

Second, to explore the robustness to the measure of GPA and the potential nonlinear relationship between GPA and wages, we classify GPA into 3 levels to explore whether different levels have different effects on wages. Namely, the "medium" level when GPA is between 2 and 3; the "good" level when GPA is between 3 and 3.5; and "excellent" level when GPA is above 3.5. Column (4) of Table 3 present the regression results. They suggest that with a good GPA, the students' starting and current wages will be 0.182 and 0.131 log points (20.0 and 14.0 percent) higher than those with medium GPA on average, respectively; and compared to the medium GPA, the premium of excellent GPA in the starting and current wages are 0.312 and 0.267 log points (36.6 and 30.6 percent) in the starting and current wages on average, respectively. In the bottom row, we also report the difference between excellent and good GPAs. It is suggested that the effects of GPA on starting wages decrease from medium to good compared with from good to excellent, while the effect of GPA is basically linear on current wage.

**Table 3. The robustness checks.**

| Variables | Controlling for different FEs | | | Alternative GPA measure | 2009 grade sample | 2020 grade sample | 2020 grade sample controlling for NCEE score | Excluding individuals from Province S |
|---|---|---|---|---|---|---|---|---|
| | (1) | (2) | (3) | (4) | (5) | (6) | (7) | (8) |
| **Panel A: Log of starting wage** | | | | | | | | |
| GPA | 0.267*** | 0.257*** | 0.260*** | | 0.239*** | 0.259*** | 0.241** | 0.259*** |
| | (0.048) | (0.041) | (0.049) | | (0.076) | (0.076) | (0.119) | (0.056) |
| Good | | | | 0.182*** | | | | |
| | | | | (0.052) | | | | |
| Excellent | | | | 0.312*** | | | | |
| | | | | (0.046) | | | | |
| R-squared | 0.411 | 0.422 | 0.433 | 0.404 | 0.423 | 0.442 | 0.196 | 0.452 |
| Excellent—good | | | | 0.130*** | | | | |
| **Panel B: Log of current wage** | | | | | | | | |
| GPA | 0.209*** | 0.230*** | 0.195*** | | 0.166 | 0.212*** | 0.255** | 0.231*** |
| | (0.062) | (0.065) | (0.063) | | (0.158) | (0.058) | (0.126) | (0.076) |
| Good | | | | 0.131** | | | | |
| | | | | (0.049) | | | | |
| Excellent | | | | 0.267*** | | | | |
| | | | | (0.068) | | | | |
| R-squared | 0.444 | 0.458 | 0.465 | 0.428 | 0.499 | 0.468 | 0.231 | 0.520 |
| Excellent—good | | | | 0.136*** | | | | |
| Controls | Yes | Yes | Yes | Yes | Yes | Yes | Yes | Yes |
| Grade FEs | No | No | No | Yes | No | No | No | Yes |
| Home province FEs | Yes | No | No | Yes | Yes | Yes | Yes | Yes |
| Currently residential FEs | No | Yes | No | Yes | Yes | Yes | Yes | Yes |
| Grade-Home province FEs | No | Yes | Yes | No | No | No | No | No |
| Grade-Currently presidential province FEs | Yes | No | Yes | No | No | No | No | No |
| Average NCEE test score | No | No | No | No | No | No | Yes | No |
| Observations | 637 | 644 | 633 | 643 | 284 | 349 | 224 | 438 |

Note: All the regressions have the same control variables as in Columns (3) and (6) in Table 1 except that Columns (1)-(6) control for different fixed effects and Column (6) additional controls for the average NCEE test scores among the students from the same province for different schools, which reduces the sample size to 224. Regressions in Columns (4) have use different measure of GPA. Specifically, we classify GPA into 3 levels: Medium– 2 to 3, good– 3 to 3.5, and excellent–above 3.5. The medium level is omitted in the regressions. Robust standard errors are in parentheses, which are clustered at the school level.

*** $p<0.01$,

** $p<0.05$,

* $p<0.1$.

Third, we try to control for students' endowment, or the knowledge and ability obtained before they entered university. To this end, we first run regressions for the grades 2009 and 2010 samples separately, which are presented in Columns (5) and (6) in Table 3. This is actually very similar with the models in Column (2), but we allow the effects of GPA and other variables to differ between the two grades. Because the number of students admitted in each province is very limited and the majors in this university are very close, regressions for different years separately with controlling for provincial fixed effects can absorb the endowment differences among different provinces to a large extent. It is shown that the results do not change

much with an exception that the effect on current wage for the 2009 grade sample reduces a little bit and become insignificant.

Then we further control for the average NCEE test score for each province, which is different for Science and Art/Social Sciences Streams. Because we only have this information for the 2010 grade sample, this makes the sample size reduce to 224. The results after controlling for these test score remain unchanged basically.

Finally, we exclude the sample from Province S. Because the university is in Province S, every year many more students are admitted from Province S than other provinces. That means, it is likely that individuals NCEE test score is far away from the average. Nevertheless, excluding these observations do not alter the results.

All in all, our results are robust to the model specification, functional form and sample composition.

## The distributional effects of GPA on wage

In previous analyses, we have illustrated the average effects of GPA on students' starting wages and their current wages. However, the effects may also vary over the distribution of students' wages. To investigate this kind of heterogeneity, we use unconditional quantile regression [28]. Unconditional quantile regression (UQR) is different from Conditional quantile regression (CQR) [29]. CQR considers the effect of GPA (in our case) on the wages at different quantiles within the "group", where the group consists of workers who share the same values of the covariates (other than GPA); while UQR consider the effect on the marginal or unconditional distribution of wages. Compared to the results of CQR, the results of UQR are more relevant with income distribution and easier to understand. The S1 Appendix provides a brief introduction to the unconditional quantile regression approach.

Fig 2 plots the fitted lines of GPA using OLS and the unconditional quantile regressions in five different quantiles of wage, 0.10th, 0.25th, 0.50th, 0.75th, and 0.90th quantiles–the left part in

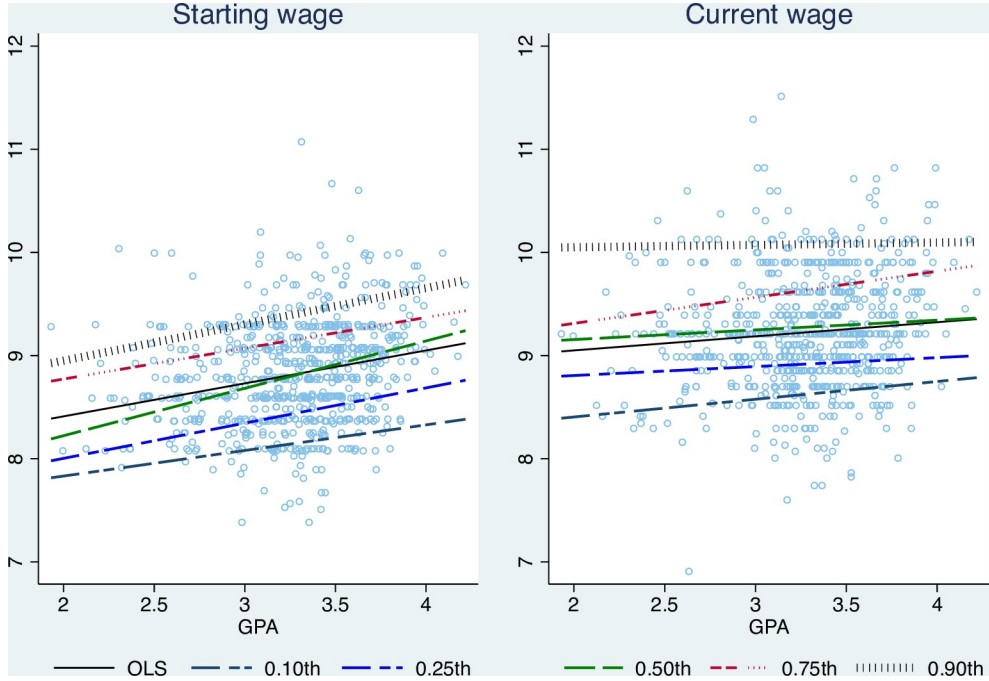

**Fig 2. The fitted regression line of different wage quantiles and OLS.**

Fig 2 presents the starting monthly wage, and the right part presents the current monthly wage. In the figure, the solid black line indicates the results of OLS, and the dot, dash, and dot-dash lines present the unconditional quantile regression results. The figure shows that the OLS regressions have different slopes with the unconditional regression lines in five quantiles. In addition, the slopes of the five quantile regression lines are also different.

To further investigate the heterogeneity of the effects over wage distributions, we estimate the effects of GPA at 19 quantiles—the 0.05th to 95th quantiles. These results are reported in Fig 3. The left part in Fig 3 shows the distributional results on the starting wage. First, GPA has a positive and significant effect on the starting wage over the whole distribution. Second, the effect is basically the same from the 0.05th to 0.80th quantiles, while it increases substantially from the 0.08th to 0.95th quantiles. This implies that GPA plays a more important role in students with higher starting wages.

The effect of the current monthly wage is presented in the right part in Fig 3. First, it is an inverted U shape with a peak in the 0.75th and 0.80th quantiles, and the coefficient of GPA is approximately 0.6. Second, compared to the left part, the effect of GPA is generally smaller in most quantiles and even insignificant from the 0.25th to 0.40th quantiles. The smaller effect on current wages may be because that when students first enter the labour market, GPA as a proxy variable of ability can help recruiters identify prospective employees [14]. Therefore, the GPA plays a greater role when the student's starting salary is higher. For current wages, after students enter the labour market, the singal effect of GPA disappears and it only serves as an index for human capital. This is why the effects on current wages are generally smaller. With respect to the effect at the 0.75th and 0.80th quantiles, GPA may interact with the job training or other factors, which causes the largest impact in these quantiles.

In general, GPA does affect students' starting monthly wages and current monthly wages, and a positive effect exists in almost all locations over the wage distribution. GPA has larger

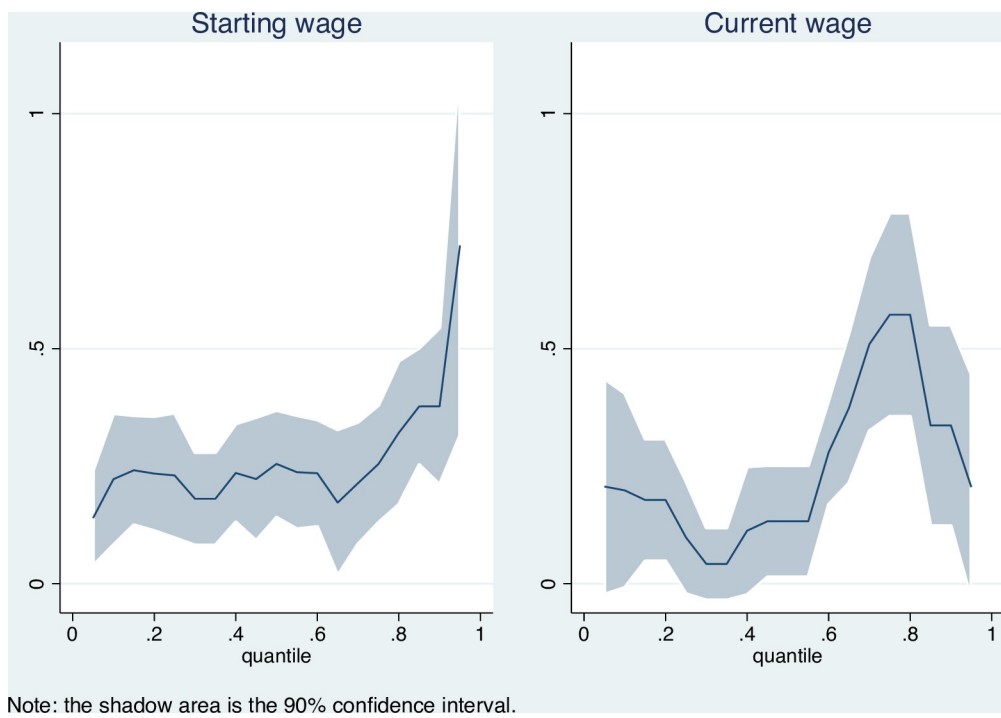

Note: the shadow area is the 90% confidence interval.

**Fig 3. The distributional effects of GPA on monthly wage.**

effects from the $0.80^{th}$ to $0.95^{th}$ quantiles on the starting wage, while the distributional effect of GPA on the current monthly wage is a U shape from the $0.05^{th}$ to $0.60^{th}$ quantile, and then becomes an inverse-U shape with peaks at the $0.75^{th}$ and $0.80^{th}$ quantiles where the effect is 82.2 percent.

## Conclusion

The relationship between student performance at school and labour market outcomes helps to understand education policy. In the past, education policy mainly focused on promoting higher education as a means of massification of education. Since 1999, China has implemented a policy of higher education expansion. From Ministry of Education of China, the gross enrolment rate increased from 12.5% in 2000 to 54.4% in 2020, nearly a fivefold increase over 20 years [30]. When degree inflation occurs due to this higher education expansion, it seems that society as a whole ignores the role of academic performance in universities. Similar to the degree, academic performance in universities should also reflect human capital accumulation and provide a signal about an individual's ability when students enter the labour market.

This study is based on data from an economics and finance university of Project 211. We find that there is a positive and significant relationship between GPA and wages. First, in the baseline results, when GPA increases by 1, starting monthly wages increase by 29.6 percent, and the effect of GPA on current monthly wages is slightly smaller than that on starting monthly wages. The results are robust. Second, in the distributional analysis, we can see that the positive effects of GPA on both wages are significant for almost all quantiles. The effect remains similar from the $0.05^{th}$ to $0.80^{th}$ quantiles and then rises as wages increase. The effect on current wage is a U shape from the $0.05^{th}$ to $0.60^{th}$ quantile, and then becomes an inverse-U shape with peaks at the $0.75^{th}$ and $0.80^{th}$ quantiles where the size of the effect is 82.2 percent. A comparison between the effects on starting wages and current wages suggests that higher GPAs, as a measure of human capital, can mainly cause higher wages; the signal effect (sheepskin effect) exists when students enter the labour market for the first time, but this effect is much smaller than the human capital effect. Regarding the specific measurement of the two roles, we need more sufficient data and more comprehensive methods to solve them, and we hope that our future research can explain them.

## Supporting information

**S1 Appendix. The unconditional quantile regression.**
(DOCX)

**S1 File.**
(DTA)

## Author Contributions

**Data curation:** Tao Zou.

**Formal analysis:** Yue Zhang.

**Investigation:** Tao Zou, Yue Zhang.

**Methodology:** Yue Zhang.

**Project administration:** Tao Zou.

**Resources:** Tao Zou, Bo Zhou.

**Software:** Yue Zhang.

**Supervision:** Tao Zou.

**Validation:** Bo Zhou.

**Visualization:** Bo Zhou.

**Writing – original draft:** Tao Zou, Yue Zhang.

**Writing – review & editing:** Tao Zou, Bo Zhou.

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
