## [Decision Letter · Decision Letter 0]

10 Mar 2022

PONE-D-22-03813Does high GPA predict or cause high wage? New evidence revisitedPLOS ONE

Dear Dr. Zou,

Thank you for submitting your manuscript to PLOS ONE. After careful consideration, we feel that it has merit but does not fully meet PLOS ONE’s publication criteria as it currently stands. Therefore, we invite you to submit a revised version of the manuscript that addresses the points raised during the review process.

Both the qualified reviewers think your paper is interesting but can be much improved. They provided useful comments. Both worry about the omitted students’ ability. If you have the college entrance exam score, it would be very helpful to include it in the model as a proxy for unobserved ability. Please try to address their concerns as much as you can. I have read your paper carefully and have a few additional comments.

You should provide a clear explanation on the relation between GPA as signal or as human capital and wages in the Introduction (and even in the abstract). For example, if GPA serves as a signal, how would starting (current) wage change with GPA? Similarly, if GPA measures human capital, how would starting (current) wage move with GPA? That is, the discussion between line 237 and 245 can be introduced early in the Introduction. You may also develop one or two testable hypotheses if this makes the writing easier.Line 27: employers’ productivity should be employees’ productivity?Can you also translate “log points” to percent change or the level change (evaluated at the sample mean) in wage? This will help readers understand how economically significant the GPA effect is.Line 64: inverted N shape is still N shape?Line 286, missing “in Table 3” in the sentence.Although I have no problem understanding your writing, there are many grammatical errors and the writing can be much improved. I suggest you hire a professional copy editor to polish your new version.Please submit your revised manuscript by Apr 24 2022 11:59PM. If you will need more time than this to complete your revisions, please reply to this message or contact the journal office at plosone@plos.org. Please include the following items when submitting your revised manuscript:A rebuttal letter that responds to each point raised by the academic editor and reviewer(s). You should upload this letter as a separate file labeled 'Response to Reviewers'.A marked-up copy of your manuscript that highlights changes made to the original version. You should upload this as a separate file labeled 'Revised Manuscript with Track Changes'.An unmarked version of your revised paper without tracked changes. You should upload this as a separate file labeled 'Manuscript'.

We look forward to receiving your revised manuscript.

Kind regards,

Shihe Fu, Ph.D.

Academic Editor

PLOS ONE

Journal Requirements:

Reviewers' comments:

Reviewer's Responses to Questions

**Comments to the Author**

1. Is the manuscript technically sound, and do the data support the conclusions?

Reviewer #1: Yes

Reviewer #2: Partly

2. Has the statistical analysis been performed appropriately and rigorously? 

Reviewer #1: Yes

Reviewer #2: No

3. Have the authors made all data underlying the findings in their manuscript fully available?

Reviewer #1: Yes

Reviewer #2: No

4. Is the manuscript presented in an intelligible fashion and written in standard English?

Reviewer #1: Yes

Reviewer #2: Yes

5. Review Comments to the Author

Reviewer #1: Referee report on:

Does high GPA predict or cause high wage? New evidence revisited

This paper uses both administrative and survey data from one elite university to examine the effect of GPA on graduates’ wages. The author(s) find that higher GPA leads to higher starting wages and wages 3-5 years after graduation. This paper contributes to the literature by using unique data from one university and by empirically estimating the effect of GPA on college graduates’ wages.

However, the paper has a great potential to be improved. Here are my comments:

1. A major issue of this research is that GPA may be endogenous and the discussion of the endogeneity issue can be strengthened. In addition, GPA may be closely related to the scores of college entrance examinations (CEE) and reflect a student’s academic and non-academic achievement before college education. Thus, GPA may not be a good measure of value added of the college education. Is it possible for the paper to include the CEE scores?

2. Data description. How were the 1000 graduates randomly selected? Is there a selection problem that 706 effective questionnaires were collected?

3. Why do the authors use unconditional quantile regression (UQR) instead of the standard (conditional) quantile regression? Are interested in the effect of GPA on the wage distribution (inequality) of college graduates? What are the pros and cons of both methods? It will also be useful for the authors to briefly introduce the UQR method.

4. The language needs to be improved. Just take the Abstract for example: “almost” � better to use “mostly”.

5. “inverse-N shape” is hard to imagine for the readers.

6. The author(s) may consider changing the title. Why use “predict or cause”? The aim of this paper seems not to distinguish between the prediction role and the causal effect of GPA.

7. Relative to its contribution to the literature, the draft is too long. The data description can be shortened.

Reviewer #2: This paper studies the relationship between students’ GPA in college and their latter job market outcome as measured by their starting wages and current wages. The data is unique.

The paper needs to clarify what economic question it tries to answer. According to the title and the contents, I think four questions are mentioned but none of them is fully addressed.

(1) Does undergraduate GPA improves the prediction of one’s wages?

If it is a prediction problem the paper aims to answer, then the paper needs to show the improvement in prediction power with or without GPA (at least say changes in R-squared). Based on the contents of the paper, I don’t think the authors try to address such a prediction problem, but then the title is misleading.

(2) Does undergraduate GPA causes higher wages? (Through screening in the job-search process?)

(3) Are knowledge learned in college (as proxied by GPA) increases one’s human capital, and in turn increases one’s latter income?

(4) How to decompose the positive relationship between GPA and wages into the above three components? (The paper title seems to propose this question, but not fully done in the paper.)

(2) and (3) are both causality problem, with different interpretation, and also different policy implications. It is a classical labor economic research question to tell apart signaling from human capital accumulation in the role college education plays to increase income. So if this paper can tell these two apart, it will be a good contribution to the literature. But the authors need to address the selection problem to answer (2) or (3). For example, students who got high GPA might make more efforts, might be more discplined, might communicate with teachers and classmate better, may love their major so that they will continue working in what they are trained for (major and occupation match). All these will contribute to both GPA and latter job market outcome.

The paper needs to be more careful with its interpretations of results. For example, it states that “high GPA can cause high wage”, does it implies that if a university increases everyone’s GPA, their wages will increase? How much of high GPA is human capital, and how much of it is personal traits?

“The comparison between the starting and the current wage suggest that the screening effect of GPA should be trivial”. This reasoning should be subject to more careful discussions. The empirical results are highly consistent with an alternative scenario, where students who are disciplined (a personal trait, a personal fixed effect, this is just one of many possibilities) got all of the three---high GPA, high starting wage, and high current wage. And this points to a selection story, not a human capital causes high wage story.

6. PLOS authors have the option to publish the peer review history of their article (what does this mean?). If published, this will include your full peer review and any attached files.

Reviewer #1: No

Reviewer #2: No

---

## [Author Response · Author response to Decision Letter 0]

23 Mar 2022

Dear editor and reviewers:

We would like to thank the editor and the reviewers for their close reading of our paper, astute comments and helpful suggestions. According to them, we carefully revised the manuscript and explained how we have responded in the revised manuscript. We also revised the format of the manuscript according to PLOS ONE’s style requirements. The major revised parts of the manuscript are marked red. Below we use two different fonts to distinguish the comments and our item-to-item responses. The figure files are processed by the Preflight Analysis and Conversion Engine (PACE), and submitted them as separated files.

PONE-D-22-03813

Does high GPA predict or cause high wage? New evidence revisited

PLOS ONE

Dear Dr. Zou,

Thank you for submitting your manuscript to PLOS ONE. After careful consideration, we feel that it has merit but does not fully meet PLOS ONE’s publication criteria as it currently stands. Therefore, we invite you to submit a revised version of the manuscript that addresses the points raised during the review process.

Both the qualified reviewers think your paper is interesting but can be much improved. They provided useful comments. Both worry about the omitted students’ ability. If you have the college entrance exam score, it would be very helpful to include it in the model as a proxy for unobserved ability. Please try to address their concerns as much as you can. I have read your paper carefully and have a few additional comments.

We totally agree that the national college entrance exam (NCEE) test score should be a good proxy variable of the unobserved ability. Unfortunately, we do not have individual’s NCEE test score. However, we have the average NCEE test score for the sciences and arts/social sciences streams separately among the students who were admitted from the particular province and year, namely the grade-home province average NCEE test score. Because the most of majors in this university we study are business and economics related, and the admission size is quite small in all provinces, the NCEE test score is very condensed. Or in other words, the NCEE test scores are very close among the students from the same province. This means that we can alleviate the concerns of the omitted unobserved ability to a large extent by controlling for this average NCEE test score, if we cannot totally eliminate this concern. To sum up, we did a series of robustness checks in lines 330-348, and columns (5)-(8) in Table 3.

We have now addressed all the referees’ questions and provided point-by-point response. We hope these revisions and responses are satisfying.

1.You should provide a clear explanation on the relation between GPA as signal or as human capital and wages in the Introduction (and even in the abstract). For example, if GPA serves as a signal, how would starting (current) wage change with GPA? Similarly, if GPA measures human capital, how would starting (current) wage move with GPA? That is, the discussion between line 237 and 245 can be introduced early in the Introduction. You may also develop one or two testable hypotheses if this makes the writing easier.

According to this suggestion, we did the following revisions in the abstract:

“Theoretically, the GPA matters for the wages due to both the human capital or signaling effect. Given that the signaling effect should diminish over time, and the effect on starting wage is a little larger than that on current wage, it is suggested that signaling effect of GPA should be trivial, and high GPA is associated with high wage should be mainly due to the human capital effect.”

In addition, we also emphasize this point in the Introduction section (lines 21-32):

“Theoretically, the impact on income is mainly through the human capital effect [6] and signal effect [7]. The human capital effect suggests that higher academic performance such as obtaining an education degree leads to greater personal productivity, thereby increasing the income of workers. The signal effect, also called sheepskin effect in the literature, believes that academic performance as a signal of labour productivity can help distinguish it from employees’ productivity. When individuals work in particular firms longer and longer, the employers can directly observe the true difference in productivity among workers. As a result, the signal effect of academic performance should diminish over time; in contrast, if the academic performance matters mainly due to the human capital effect, the effect of academic performance should not reduce substantially and even the effect can increase over time [8-9].”

2.Line 27: employers’ productivity should be employees’ productivity?

 Thank you for pointing this out. We have modified this mistake.

3.Can you also translate “log points” to percent change or the level change (evaluated at the sample mean) in wage? This will help readers understand how economically significant the GPA effect is.

Thank you for your significant reminding. We have translated all “log point” to percentage change in this article. For example, in the abstract and Introduction, we changed the “As GPA increases by 1 unit, the starting monthly wage goes up by 0.259 log points on average” to “As GPA increases by 1 unit, the starting monthly wage goes up by 29.6 percent on average”; in the section of empirical analyses, we change the expression to the following style: “from 0.278 log points (32.05 percent) to 0.225 log points (25.23 percent).” Here we report both the changes in log points and percentage. Reporting the former one is to keep consistent with the results presented in tables and reporting the latter is to make it easy to understand the economic significance.

4.Line 64: inverted N shape is still N shape?

Inverted N shape is not N shape, but we agree that it is not easy to understand. Therefore, we rephrased the expression as the following one (lines 64-66):

“For current wage, it is a U shape from the 0.05th to 0.60th quantile, and from then onward becomes an inverse-U shape with a peak at the 0.75th and 0.80th quantiles where the effect is 82.2 percent when GPA increases by one unit.”

5.Line 286, missing “in Table 3” in the sentence.

Thank you for this suggestion. We have added it in line 295.

6.Although I have no problem understanding your writing, there are many grammatical errors and the writing can be much improved. I suggest you hire a professional copy editor to polish your new version.

Thank you for your suggestion. We have hired a professional editor to polished the writing.

A rebuttal letter that responds to each point raised by the academic editor and reviewer(s). You should upload this letter as a separate file labeled 'Response to Reviewers'.

A marked-up copy of your manuscript that highlights changes made to the original version. You should upload this as a separate file labeled 'Revised Manuscript with Track Changes'.

An unmarked version of your revised paper without tracked changes. You should upload this as a separate file labeled 'Manuscript'.

 Yes, we have attached all of these files when resubmitted this article.

We look forward to receiving your revised manuscript.

Kind regards,

Shihe Fu, Ph.D.

Academic Editor

PLOS ONE

Journal Requirements:

Yes, we have changed the paper format according to the PLOS ONE’s style.

We have added this ethic statement in the ‘Method’ section: lines 113-131.

Reviewers' comments:

Reviewer's Responses to Questions

Comments to the Author

1. Is the manuscript technically sound, and do the data support the conclusions?

Reviewer #1: Yes

Reviewer #2: Partly

2. Has the statistical analysis been performed appropriately and rigorously?

Reviewer #1: Yes

Reviewer #2: No

3. Have the authors made all data underlying the findings in their manuscript fully available?

Reviewer #1: Yes

Reviewer #2: No

4. Is the manuscript presented in an intelligible fashion and written in standard English?

Reviewer #1: Yes

Reviewer #2: Yes

5. Review Comments to the Author

Reviewer #1: Referee report on:

Does high GPA predict or cause high wage? New evidence revisited

This paper uses both administrative and survey data from one elite university to examine the effect of GPA on graduates’ wages. The author(s) find that higher GPA leads to higher starting wages and wages 3-5 years after graduation. This paper contributes to the literature by using unique data from one university and by empirically estimating the effect of GPA on college graduates’ wages.

However, the paper has a great potential to be improved. Here are my comments:

1. A major issue of this research is that GPA may be endogenous and the discussion of the endogeneity issue can be strengthened. In addition, GPA may be closely related to the scores of college entrance examinations (CEE) and reflect a student’s academic and non-academic achievement before college education. Thus, GPA may not be a good measure of value added of the college education. Is it possible for the paper to include the CEE scores?

We agree that if the regression includes CEE scores, the endogeneity issue will be largely alleviated. Unfortunately, we do not have individual’s CEE score. To alleviate the endogeneity concern as possible as we can, we did a series of robustness checks (lines 330-348, and columns (5)-(8) in Table 3). First, we control for the Home Province FE and Grade FE to approximately control for students’ CEE scores. Because universities enrolment was organized in each province, while the enrolment number of this university is not large every year in each province except in Province S where the university is, and the most majors in this university are business and economics related, the CEE test scores students from the same province every year are quite close. Thus, controlling for these two fixed effects can largely capture individual’s ability. Second, we run the regressions for the Grades 2009 and 2000 separately. This is actually very similar with the above test – controlling for Home Province FE, but more flexible, because we allow the effects of GPA and other variables can change over time. Third, although we do not know individual’s CEE test score, we know the average CEE test for Science and Art/Social Sciences streams in each province and year. Controlling for these test scores can further alleviate the endogeneity concern. Lastly, we exclude the sample from Province S, where many more students than other provinces were admitted. Excluding these sample makes the average CEE test score are more relevant with individual’s test score. The regression results in all of these robustness checks are close to our baseline results.

We hope the reviewer believes these efforts can more or less alleviate the endogeneity concern.

2. Data description. How were the 1000 graduates randomly selected? Is there a selection problem that 706 effective questionnaires were collected?

The university has a roaster for all of the university graduates, which includes some “permanent” contact information, such as email and QQ (a popular instant massage application in China) account. This roaster is the sampling frame. From the roaster, 1000 graduates were randomly selected. Regarding the selection bias, we believe there should be. Usually the most and least successful graduates are less likely to respond the survey, because they are either too busy or reluctant to report their ‘frustrated’ situation. This may cause some sample selection bias, but we cannot investigate more on this due to the data limitation. We added a footnote to admitted this potential problem (footnote 7 in Page 5):

“The university has a roaster of graduates, which includes some ‘permanent’ contact information, such as email and QQ account (a popular instant massage application in China). 1000 graduates were randomly selected from this roaster. The most and least successful graduates are less likely to respond the survey, because they are either too busy to fill in the survey form or reluctant to report their ‘frustrated’ situation. This may cause some bias, but we cannot investigate more due to the data limitation.”

3. Why do the authors use unconditional quantile regression (UQR) instead of the standard (conditional) quantile regression? Are interested in the effect of GPA on the wage distribution (inequality) of college graduates? What are the pros and cons of both methods? It will also be useful for the authors to briefly introduce the UQR method.

Unconditional quantile regression and conditional quantile regression answers different questions. Conditional quantile regression considers the effect of GPA (in our case) on the wages at different quantiles within the “group”, where the groups consists of workers who share the same values of the covariates (other than GPA); while unconditional quantile regression consider the effect on the marginal or unconditional distribution of wages. Because the marginal distribution is more relevant with income distribution and the estimation results of unconditional quantile regression is easier to understand than the conditional quantile regression especially when covariates are continuous (Firpo et al., 2009), more and more studies adopt the unconditional quantile regression approach. In the revised manuscript, we added a footnote (footnote 13 in Page 21) to discuss this:

“Unconditional quantile regression (UQR) is different from Conditional quantile regression (CQR) [25]. CQR considers the effect of GPA (in our case) on the wages at different quantiles within the “group”, where the group consists of workers who share the same values of the covariates (other than GPA); while UQR consider the effect on the marginal or unconditional distribution of wages. Compared to the results of CQR, the results of UQR is more relevant with income distribution and easier to understand. The appendix provides a brief introduction to the unconditional quantile regression approach.”

We have a brief introduction to unconditional quantile regression in Appendix. If the reviewer believe that we should put it into the main text, we can do that.

4. The language needs to be improved. Just take the Abstract for example: “almost” � better to use “mostly”.

Thank you for your suggestion. We have polished this article.

5. “inverse-N shape” is hard to imagine for the readers.

We agree that it is not easy to understand. Therefore, we rephrased the expression as the following one (lines 64-66):

“For current wage, it is a U shape from the 0.05th to 0.60th quantile, and from then onward becomes an inverse-U shape with a peak at the 0.75th and 0.80th quantiles where the effect is 82.2 percent when GPA increases by one unit.”

6. The author(s) may consider changing the title. Why use “predict or cause”? The aim of this paper seems not to distinguish between the prediction role and the causal effect of GPA.

We have changed the title to “Does GPA matter for university graduates’ wages? New evidence revisited.”

7. Relative to its contribution to the literature, the draft is too long. The data description can be shortened.

We agree with the referee. We have shortened the data description part.

Reviewer #2: This paper studies the relationship between students’ GPA in college and their latter job market outcome as measured by their starting wages and current wages. The data is unique.

The paper needs to clarify what economic question it tries to answer. According to the title and the contents, I think four questions are mentioned but none of them is fully addressed.

Many thanks to the comments of the reviewer. We carefully think of these comments and suggestions, and made a few changes to address the concerns of the reviewer. Because the four questions are related to each other, we first provide a simple response to each question and then provide a comprehensive response in the end. 

(1) Does undergraduate GPA improves the prediction of one’s wages?

If it is a prediction problem the paper aims to answer, then the paper needs to show the improvement in prediction power with or without GPA (at least say changes in R-squared). Based on the contents of the paper, I don’t think the authors try to address such a prediction problem, but then the title is misleading.

Thanks for pointing out this. We did not realize this. We think our paper aims to answer these three questions: (1) Does GPA affect university graduates’ wages? (2) Does the impact of GPA vary between starting wages and wages 3–5 years after graduation? (3) Is the effect heterogeneous over the wage distribution? This actually has nothing to do with more precise prediction. Thus, we change the title to “Does GPA matter for university graduates’ wages? New evidence revisited”.

(2) Does undergraduate GPA causes higher wages? (Through screening in the job-search process?)

See the response for the following question. We reply these two questions together.

(3) Are knowledge learned in college (as proxied by GPA) increases one’s human capital, and in turn increases one’s latter income?

We believe that GPA matters for wage both due to the screening effects and human capital, but it is hard to distinguish between them. The comparison between the effects on the starting wage and current wage may shed some light on this, but we soften our statement and do not want to over-interpret this. 

(4) How to decompose the positive relationship between GPA and wages into the above three components? (The paper title seems to propose this question, but not fully done in the paper.)

We do not think we can make it. Thus, we change the title of the paper. 

(2) and (3) are both causality problem, with different interpretation, and also different policy implications. It is a classical labor economic research question to tell apart signaling from human capital accumulation in the role college education plays to increase income. So if this paper can tell these two apart, it will be a good contribution to the literature. But the authors need to address the selection problem to answer (2) or (3). For example, students who got high GPA might make more efforts, might be more discplined, might communicate with teachers and classmate better, may love their major so that they will continue working in what they are trained for (major and occupation match). All these will contribute to both GPA and latter job market outcome.

The paper needs to be more careful with its interpretations of results. For example, it states that “high GPA can cause high wage”, does it implies that if a university increases everyone’s GPA, their wages will increase? How much of high GPA is human capital, and how much of it is personal traits?

First, we do not think we can decompose the human capital in GPA, and also distinguish human capital from personal traits. In fact, it is also believed that the personal traits are some kind of human capital, such as Heckman (2011), Heckman and Kautz (2012, 2013).

Second, we do not think university should raise everyone’s GPA, but our results show that within the same university, the high GPA is associated with high wage. We try to alleviate the endogeneity concerns as possible as we can in the robustness checks, and we believe that this should confirm some causal effect here. This result implies that it is worthwhile for college students working hard to get high GPA 

Reference: 

Heckman, J. J. (2011). The value of early childhood education. American Educator, 31,31–36.

Heckman, J. J., & Kautz, T. (2012). Hard evidence on soft skills. Labour Economics, 19, 451–464. https://doi.org/10.1016/j.labeco.2012.05.014.

Heckman, J. J., & Kautz, T. (2013). Fostering and measuring skills: Interventions that improve character and cognition. Cambridge, MA: National Bureau of Economic Research. https://doi.org/10.3386/w19656.

“The comparison between the starting and the current wage suggest that the screening effect of GPA should be trivial”. This reasoning should be subject to more careful discussions. The empirical results are highly consistent with an alternative scenario, where students who are disciplined (a personal trait, a personal fixed effect, this is just one of many possibilities) got all of the three---high GPA, high starting wage, and high current wage. And this points to a selection story, not a human capital causes high wage story.

Many thanks to these comments. Here we would like to provide some comprehensive response for the above questions.

We do not think we can clearly distinguish the human capital theory and signal effect based on our data, so we tend to describe a pattern about GPA and starting wage and later years’ wage. Based on this work, we also try to shed some light on the human capital effect and signal effect by comparing the effects of GPA on starting wage and current wage. After all, the previous studies such as Belman & Heywood (1997), Liu & Wong (1982) and Antelius (2000) have shown that the signal effect will diminish over time, while the human capital effect does not and even increase over time.

In order to further alleviate the endogeneity concerns as possible as we can, we did a series of robustness checks. First, we control for the Home Province FE and Grade FE to approximately control for students’ CEE scores. Because universities enrolment was organized in each province, while the enrolment number of this university is not large every year in each province except Province S where the university is and the most majors in this university are business and economics related, the CEE test scores students from the same province every year are quite close. Thus, controlling for these two fixed effects can largely capture individual’s ability. Second, we run the regressions for the Grades 2009 and 2000 separately. This is actually very similar with the above test – controlling for Home Province FE, but more flexible, because we allow the effects of GPA and other variables can change over time. Third, although we do not know individual’s CEE test score, we know the average CEE test for Science and Art/Social Sciences streams in each province and year. Controlling for these test scores can further alleviate the endogeneity concern. Lastly, we exclude the sample from Province S, where much more students than other provinces were admitted. This makes the average CEE test score are more relevant with individual’s test score. The regression results in all of these robustness checks are close to our baseline results.

Regarding the policy implication, we think the results show that the GPA is still very important for college students. Because GPA mainly reflect the relative academic performance in university, and GPA matters for wages, students need to pay much attention on this. Knowing this information may help some students who are at the margin between diligent studying and “enjoying” university life.

6. PLOS authors have the option to publish the peer review history of their article (what does this mean?). If published, this will include your full peer review and any attached files.

Do you want your identity to be public for this peer review? For information about this choice, including consent withdrawal, please see our Privacy Policy.

Reviewer #1: No

Reviewer #2: No

Yes, we have use PACE to process the figures.

---

## [Decision Letter · Decision Letter 1]

31 Mar 2022

Does GPA matter for university graduates’ wages? New evidence revisited

PONE-D-22-03813R1

Dear Dr. Zou,

Both the reviewers are happy with your revision and recommended "accept." We’re pleased to inform you that your manuscript has been judged scientifically suitable for publication and will be formally accepted for publication once it meets all outstanding technical requirements. Please note that there are a few minior grammatical errors or typos in your paper, so please make sure you can correct them during the proofreading step. Also, I think there is no need to paste the ethical approval letters in the paper; you can just include a summary statement following the PLOS ONE style.

Kind regards,

Shihe Fu, Ph.D.

Academic Editor

PLOS ONE

Additional Editor Comments (optional):

Reviewers' comments:

Reviewer's Responses to Questions

**Comments to the Author**

1. If the authors have adequately addressed your comments raised in a previous round of review and you feel that this manuscript is now acceptable for publication, you may indicate that here to bypass the “Comments to the Author” section, enter your conflict of interest statement in the “Confidential to Editor” section, and submit your "Accept" recommendation.

Reviewer #1: All comments have been addressed

Reviewer #2: All comments have been addressed

2. Is the manuscript technically sound, and do the data support the conclusions?

Reviewer #1: Yes

Reviewer #2: Partly

3. Has the statistical analysis been performed appropriately and rigorously? 

Reviewer #1: Yes

Reviewer #2: Yes

4. Have the authors made all data underlying the findings in their manuscript fully available?

Reviewer #1: Yes

Reviewer #2: Yes

5. Is the manuscript presented in an intelligible fashion and written in standard English?

Reviewer #1: Yes

Reviewer #2: Yes

6. Review Comments to the Author

Reviewer #1: The authors have made significant changes, and the paper quality has improved significantly. However, the language can still be improved. Take the Abstract, for example.

1. "Students are homogenous since their majors are closely related to economics and business The OLS regression results indicate......" There should be a period between these two sentences.

2. "the GPA matters for the wages due to both the human capital or signaling effect." The grammar seems incorrect.

I suggest that the authors have a thorough check of the language of this manuscript.

Reviewer #2: The province-year level average of NCEE scores may not be comparable across provinces, as different provinces use different textbooks/exams. So I wonder whether this is a good way to control for ability. Another issue regarding "selection" is the diligence/being discipline/compliance. I understand that the authors don't have further data to address this kind of things. But, as long as their interpretations of the results are aligned with their evidence, I think it is OK.

7. PLOS authors have the option to publish the peer review history of their article (what does this mean?). If published, this will include your full peer review and any attached files.

Reviewer #1: No

Reviewer #2: No

---

## [Editor Report · Acceptance letter]

4 Apr 2022

PONE-D-22-03813R1 

Does GPA matter for university graduates’ wages?
New evidence revisited 

Dear Dr. Zou:

I'm pleased to inform you that your manuscript has been deemed suitable for publication in PLOS ONE. Congratulations! Your manuscript is now with our production department. 

Kind regards, 

on behalf of

Dr. Shihe Fu 

Academic Editor

PLOS ONE